# Identification of Microorganisms from Several Surfaces by MALDI-TOF MS: *P. aeruginosa* Is Leading in Biofilm Formation

**DOI:** 10.3390/microorganisms9050992

**Published:** 2021-05-04

**Authors:** Ehsan Asghari, Annika Kiel, Bernhard Peter Kaltschmidt, Martin Wortmann, Nadine Schmidt, Bruno Hüsgen, Andreas Hütten, Cornelius Knabbe, Christian Kaltschmidt, Barbara Kaltschmidt

**Affiliations:** 1Department of Cell Biology, Faculty of Biology, University of Bielefeld, 33615 Bielefeld, Germany; ehsan.asghari@uni-bielefeld.de (E.A.); Annika.kiel@uni-bielefeld.de (A.K.); C.Kaltschmidt@uni-bielefeld.de (C.K.); 2Department of Thin Films & Physics of Nanostructures, Center of Spinelectronic Materials and Devices, Faculty of Physics, University of Bielefeld, 33615 Bielefeld, Germany; b.kaltschmidt@uni-bielefeld.de (B.P.K.); Andreas.huetten@uni-bielefeld.de (A.H.); 3Department of Plastics Technology, University of Applied Sciences, 33619 Bielefeld, Germany; martin.wortmann@fh-bielefeld.de (M.W.); bruno.huesgen@fh-bielefeld.de (B.H.); 4Institute for Laboratory- and Transfusion Medicine, Heart- and Diabetes Centre NRW, Ruhr-University Bochum, 32545 Bad Oeynhausen, Germany; naschmidt@hdz-nrw.de (N.S.); cknabbe@hdz-nrw.de (C.K.)

**Keywords:** domestic appliances, biofilm, MALDI-TOF-MS, scanning electron microscopy, *Pseudomonas aeruginosa*

## Abstract

New ecological trends and changes in consumer behavior are known to favor biofilm formation in household appliances, increasing the need for new antimicrobial materials and surfaces. Their development requires laboratory-cultivated biofilms, or biofilm model systems (BMS), which allow for accelerated growth and offer better understanding of the underlying formation mechanisms. Here, we identified bacterial strains in wildtype biofilms from a variety of materials from domestic appliances using matrix-assisted laser desorption/ionization-time of flight mass spectroscopy (MALDI-TOF-MS). Staphylococci and pseudomonads were identified by MALDI-TOF-MS as the main genera in the habitats and were analyzed for biofilm formation using various in vitro methods. Standard quantitative biofilm assays were combined with scanning electron microscopy (SEM) to characterize biofilm formation. While *P**seudomonas putida*, a published lead germ, was not identified in any of the collected samples, *Pseudomonas aeruginosa* was found to be the most dominant biofilm producer. Water-born *Pseudomonads* were dominantly found in compartments with water contact only, such as in detergent compartment and detergent enemata. Furthermore, materials in contact with the washing load are predominantly colonized with bacteria from the human.

## 1. Introduction

Over the last decade, the transition towards a more sustainable society has led to significant changes in consumer behavior. Amongst the most predominant trends are preservation of water and energy and an increased use of natural, environmentally benign materials and chemicals. These changes have led to lower operating temperatures and reduced water consumption in appliances as well as to an increased use of pH-neutral, bleach-free, and/or biodegradable detergents and cleaning agents [1,2,3]. While being effective in reducing our environmental footprint, these ecological trends also bear some less desirable consequences. In recent years, several studies reported on the formation of biofilms in home environments [4] and household appliances such as washing machines [5,6], dishwashers [7], coffee makers [8], and food-processing equipment [9]. During operation, microorganisms are introduced by contaminated tap water [6] or the dirty load (dishes, laundry etc.) [5] and, if environmental conditions are favorable, can lead to the growth of bacteria on internal and external surface. This bacteria accumulation, or cohabitation, commonly referred to as biofilm, comprises microorganisms in their own ecosystem [10], which can stick to almost any surface in an aqueous environment [11]. 

While these microorganisms can be found on many materials and surfaces in homes, fully developed biofilms can lead to malodor [12], corrosion or even clogging of pipes and drains [13,14,15]. To counteract these trends, appliance manufacturers have put in place instructions for cleaning and maintenance and developed special cleaning programs to prevent formation of biofilms. However, as the current ecological trends are expected to continue, a more proactive approach to biofilm prevention may be needed. In order to inhibit biofilm formation without the use of external cleaners and environmentally harmful chemicals, more advanced materials and engineered surfaces with antimicrobial properties must be developed and become an integral part of product design. This in turn requires a better understanding of the mechanisms of biofilm formation and its dependence on environmental factors and surface properties. 

A biofilm is a structured, organized community with functional heterogeneity, forming a self-produced extracellular polymeric substance (EPS) matrix. The EPS matrix contains components of lysed bacteria, carbohydrates, proteins, nucleic acids and humic substances and consists of a complex structure with channels for nutrient flow [16]. Biofilms enable bacteria to defend themselves against external threats, for example, by exhibiting antibiotic resistance or tolerance towards detergents and disinfectants [17,18,19]. The sessile form also allows bacteria to quickly adapt to changes in their environment, as the EPS matrix is a highly dynamic ecosystem that is continuously remodeled [20,21,22,23]. While most biofilms found in household environments share the above characteristics, they can differ substantially with respect to nature and quantity of the bacterial species contained in the films. Even within a single house appliance, the composition of the biofilm has been found to be different for different sampling locations, suggesting a strong sensitivity towards environment factors and material/surface properties [24].

The formation of biofilms occurs in several stages: (1) attachment of bacteria and adhesion to the surface, (2) surface cover (monolayer) and production of the extracellular matrix, (3) formation of a microcolony (multilayer), (4) maturation and development of characteristic column and tower structures due to the polysaccharides, and (5) dissemination and dispersion [25]. Given its protective properties, once the EPS matrix has formed, it becomes difficult to effectively remove biofilms in areas that cannot be cleaned by mechanical force. The key to preventing biofilm growth is thus inhibiting the irreversible attachment of bacteria on the surface. This initial step is governed by the attractive and repulsive forces between the microorganisms and the surface (e.g., van der Waals forces, electrostatic forces and hydrophobic interactions [26,27,28] that are, in turn, largely dependent on the properties of the surface (roughness, micro-topography) and its physical/chemical environment (surface charge, hydrophobicity) [29]. For example, it was reported that surface scratches in a size similar to the bacterial diameter lead to the strongest bacterial colonization, whereas scratches with a larger diameter are not attractive for bacterial growth [30]. Special topographic features can render surfaces highly hydrophobic, thus repelling water and inhibiting adhesion of microbes. Using this approach, lotus leaves prevent water from wetting its surface. Droplets roll down the leaf and clean the surface by picking up bacteria, dirt and other fine debris [31]. The microstructure of shark skin comprises parallel ridges and grooves, forming a unique diamond pattern arrangement of denticles with tiny riblets, which prevent microorganisms from attaching on the shark skin [32].

The successful development of antimicrobial surfaces and materials requires a thorough understanding of how environmental factors (e.g., temperature, chemistry, humidity) affect these microbiological systems and how the tailoring of substrate materials and surface morphologies can actively inhibit biofilm formation. Unfortunately, due to the often unspecified growth conditions, slow growth rates and high complexity of real-life biofilms, the analysis of the effects of individual environmental factors on biofilm formation is challenging and inhibits complete understanding of the growth mechanism. Therefore, researchers must be able to study these biofilms in the laboratory, requiring adequate model systems that simulate real-life environments. These model systems must allow for accelerated growth rates and should comprise bacterial species that are common to biofilms found in domestic environments. In addition, such a model system likely requires new methods for bacterial cultivation under real-life conditions, using adjusted nutrient mixtures and temperature profiles.

The first step in the development of a representative model system is the selection of suitable characterization techniques that allow for unambiguous identification of the most common bacterial strains. Previous studies of bacterial ecosystems concentrated largely on classical microbiology for species determination [6] or used modern genomic classification by 16S rRNA analysis but without characterization of bacterial cultures. The classical microbiology is based on cultivation and visual inspection by microscopy. The main disadvantages of classical microbiology for characterization and classification of bacteria are the long incubation time (6 to 12 h) and the wide spectrum of metabolic properties which obfuscates easy identification. More advanced molecular genetic methods, such as 16S rRNA sequencing, allow for easier classification of bacteria, but are more difficult to standardize and complicate identification of individual species. In addition, protein-based methods are not hampered by the third wobble base of the nucleic acid triplet.

In recent years, a new laboratory standard for determination of bacterial species was established. Single colonies isolated from complex biofilms are cultivated and subsequently analyzed by matrix-assisted laser desorption/ionization-time of flight mass spectroscopy (MALDI-TOF-MS) [33]. MALD-TOF-MS can be very accurate when it comes to identifying bacterial species at a level up to 94% but, depending on the laboratory, this can drop to 42% [34]. Factors limiting general use of MALDI-TOF-MS in mixed cultures is the need for single colonies to grow on defined growth media, a procedure which we adapted in our analysis. However, this method has a potential to replace other differentiation methods in microbiology as it offers significant advantages like ease of operation, low cost, short analysis time, high sensitivity and accuracy in species identification. 

In this study, we aimed to identify bacterial species suitable for the development of a biofilm model system by combining state-of-the-art MALDI-TOF-MS for species identification with standard quantitative biofilm assays and SEM. Candidate species were extracted from naturally grown biofilm samples and cultivated in a laboratory to further investigate biofilm formation on two common materials, glass and plastic (polystyrene). 

## 2. Material and Methods

### 2.1. Sample Collection

Samples were collected from different material surfaces in six household washing machines (varying brand, model, and age) using a sterile cotton swab and plated onto Luria broth (LB) plates. Four different material categories were chosen: glass, metal (steel), elastomers (rubber), and plastic. Five different swabbing locations covering the different material categories were picked: inner site of the detergent compartment (plastic), detergent enema (plastic), washing drum (metal, steel), sealing rubber (elastomers, rubber), and the inner sight glass (glass). LB plates were incubated overnight at 37 °C. As comparative bacterial strains for our experiments, the strains *Pseudomonas putida* DSM 100120 from Deutsche Sammlung von Mikroorganismen und Zellkulturen GmbH (DSMZ) and the laboratory *Escherichia coli* XL1-blue strain (Stratagene/Agilent, San Diego, CA, USA) were included. 

### 2.2. Isolation and Identification of Bacterial Strains

For isolation of bacterial strains, four colonies from each plate harboring bacteria of the five different swabbing locations (see above) were picked randomly, plated onto new LB agar plates, and incubated overnight at 37 °C. Isolated colonies were identified using matrix-assisted laser desorption/ionization-time of flight mass spectrometry (MALDI-TOF-MS) instrument, Microflex LT bioanalyzer (Bruker, Bremen, Germany), MBT Compass software ver. 4.2, and Compass Library DB-7854. The samples were prepared by direct transfer method as described before [35]. Fresh colony material was smeared on a polished steel MSP 96 target (Bruker Daltonik) using a toothpick, overlaid with 1 μL of a saturated a-cyano-4-hydroxy-cinnamic acid (HCCA) matrix solution in 50% acetonitrile–2.5% trifluoroacetic acid (Bruker Daltonik), and air-dried at room temperature. Measurements were done with two technical replicates. The standard Bruker interpretative criteria were applied. Scores > 2 in both technical replicates were used for reliable species identification. If mass spectrometry could not clearly identify the sample, classical microbiology was used for identification as described in, e.g., Current Methods for Classification and Identification of Microorganisms [36]. These include gram-staining, catalase and oxidase measurement, as well as biochemical analysis by Vitek 2 (Biomerieux, Marcy-l’Étoile, France).

### 2.3. Detection of Biofilm Formation

#### 2.3.1. Substrate Materials and Surface Treatments

In this study, we utilized laboratory glass tubes and laboratory polystyrene culture plates to represent the common material categories glass and plastic, respectively. Biofilm assays were conducted using borosilicate glass tubes (SCHOTT AG, DURAN, Mainz, Germany), borosilicate glass coverslips (BRAND GmbH, Wertheim, Germany), polystyrene cell culture plates with a flat bottom (F-bottom, declared hydrophilic) (TPP, Transadingen, Switzerland), and mid-binding 8-strips polystyrene cell culture plates with a U-shaped bottom (U-bottom, declared hydrophobic) (Greiner, Frickenhausen, Germany). The water contact angle was measured using a contact angle goniometer OCA 15 Pro (Dataphysics, Filderstadt, Germany). The measurements were performed with 5 µL water droplets at the inner surface of the polystyrene well plates in ambient conditions. We performed eight measurements for each material. For contact angles below 90°, the surface is considered hydrophilic. Contact angles larger than 90° are considered hydrophobic [37]. In this study, we declared the F-bottom plates hydrophilic and the U-bottom plateshydrophobic.

#### 2.3.2. Congo Red Agar Method (CRA)

Freeman et al. described a qualitative assay to detect the microorganisms which are able to produce biofilm [38]. This method is based on color changing of colonies on the Congo red agar (CRA) medium. Colonies with black color represent a biofilm producer whereas red-pink colonies retain non-biofilm producers. The CRA medium comprised brain heart infusion (BHI) (37 g/L) (Carl Roth GmbH + Co. KG, Karlsruhe, Germany), sucrose (50 g/L) (Carl Roth GmbH + Co. KG, Karlsruhe, Germany), agar (10 g/L) (Carl Roth GmbH + Co. KG, Karlsruhe, Germany), and Congo red dye (0.8 g/L) (Sigma-Aldrich, St. Louis, MO, USA). Congo red dye was prepared separately and added into the autoclaved BHI medium. The plates were incubated at 37 °C for 24 h aerobically.

#### 2.3.3. Microtiter Plate Method (MtP)

Microtiter plate assay is the most common quantitative assay to detect biofilm as determined by Christensen et al. [39]. For this study, we inoculated all the bacteria from fresh agar plates into both LB and BHI media and incubated at 37 °C overnight in static conditions. The precultures were adjusted to OD_600_ = 0.1 in the fresh medium using a UV-6300PC Double Beam Spectrophotometer (VWR, Radnor, PA, USA). Ninety-six-well flat bottom cell culture plate (hydrophilic) and 8-strips U-form bottom plate (hydrophobic) were filled with 200 µL diluted culture. The plates were incubated at 37 °C for 24 h without shaking. After two washes with the phosphate-buffered saline (PBS) (pH 7.2) to remove the planktonic bacteria, the wells were stained with 0.1% crystal violet (CV). Before staining with crystal violet, the plates were dried at 60 °C for two hours. The stained biofilm was resolubilized with 225 µL of 30% acetic acid and measured spectrophotometrically at 595 nm with a PowerWave microplate reader (BioTek, Winooski, VT, USA). All the samples were tested in triplicate. If OD values were above 3, a dilution was performed. The average score of three wells was calculated. 

#### 2.3.4. Tube Method (TM)

Tube method is a quantification assay to detect biofilm formation described by Christensen et al. [40]. For the tube method, we used the same procedure as MtP with two differences. We used glass tubes instead of polystyrene test tubes and filled the tubes with 2 mL of OD_600_ = 0.1 adjusted bacterial culture. All the samples were tested in technical triplicates. Cristal violet staining was performed as previously described; volumes were adjusted to 2 mL. Photometric measurement of the stained biofilm resolubilized in 2 mL of 30% acetic acid was performed with 200 µL of the solution in a 96-well flat bottom cell culture plate at 595 nm using a PowerWave microplate reader (BioTek, Winooski, VT, USA).

#### 2.3.5. Colony-Forming Unit Method (CFU)

For direct quantification of viable cell numbers, we performed the colony-forming unit method. For this method, the precultures were inoculated from fresh agar plates as described above (MtP method) using the LB and BHI media for all idendified Staphylococci and Pseudomonads. For each bacterium, three glass tubes containing 2 mL medium, LB and BHI, were inoculated and OD_600_ = 0.1 was adjusted. Glass tubes were incubated overnight at 37 °C in static conditions. After incubation, the potential produced biofilm was washed with physiological saline (0.9% NaCl) two to three times. After that, the potential biofilm was dissolved in 5 mL physiological saline using three to four glass beads per glass tube and vigorous vortexing. Dilution series were performed (1:10, 1:100, 1:1000, 1:10,000) and 100 µL bacterial dilutions were plated onto fresh agar plates (LB and BHI). The plates were incubated overnight at 37 °C. After incubation, colonies were counted, and the colony-forming units were calculated. 

#### 2.3.6. Scanning Electron Microscopy (SEM)

To grow a biofilm, a preculture of *P. aeruginosa* was adjusted to OD_600_ = 0.1 and 1 mL was added onto glass coverslips. We incubated one sample for 0 h (negative control), 12 h and another sample for 72 h.

After 0, 12 and 72 h of incubation at 37 °C without shaking, the samples were rinsed twice with 1% PBS. The samples were fixated with Karnovsky solution (2% paraformaldehyde, 2.5% glutaraldehyde) for 30 min. The fixed samples were dehydrated using 50, 70, 80, 90, 95, and 100% (v/v) graded ethanol. The samples were sputter-coated with 4 nm ruthenium and then investigated with a Helios NanoLab DualBeam 600 (FEI Company, Hillsboro, OR, USA) scanning electron microscope (SEM) at an acceleration voltage of 2 kV and a beam current of 0.17 nA. In order to get an impression of both the top view and the side view of components of the biofilm, the samples were tilted 52° when necessary. The SEM images were colored using the Adobe Photoshop CS 3 version 10 software. The highlighted sections were selected using the quick selection tool and then colored by picking the adjustment layer: solid color option. After that, the color blend mode was chosen.

### 2.4. Statistical Analysis

For statistical analysis, a two-way ANOVA or mixed model ANOVA (due to missing datapoints) was performed for the results of the MtP, TM and CFU assays. We compared the effect of the media within each bacterial species, as well as between all bacterial species. Alpha was chosen to be 0.05 and *p*-values were set to 0.1234 (ns), 0.0332 (*), 0.0021(**), 0.0002 (***), and <0.0001 (****). Statistical significance was visualized and discussed only for high significance (***, ****). 

### 2.5. Phylogenic Analysis of 16S rRNA of the Bacterial Spectra 

For the analysis of homology of 16S rRNA sequences of pseudomonads and staphylococci, we used the multiple sequence alignment tool MegAlignPro using the bioinformatics software DNASTAR Lasergene 17 (DNASTAR, Madison, WI, USA). For the multiple sequence alignment, the ClustalW alignment tool was used. Afterwards, the Neighbor Joining BIONJ tool was used to create the phylogenic tree. The 16S rRNA sequences were taken from the PubMed library. 

## 3. Results 

### 3.1. Isolation, Cultivation, and Identification of Bacteria

The samples were collected from different material surfaces representing the common material categories glass, metal (steel), elastomers (rubber), and plastic (thermoplastic). Given the fact that other influential factors such as temperature, type and concentration of detergents and humidity levels vary substantially during operation over a long period of time (many years), explicit statements on the history and effect of these environmental factors cannot be made. In this study, we distinguish explicitly between different substrate materials and assume that all samples had contact with water containing detergent, were exposed to elevated temperatures (>30 °C) and underwent multiple wet/dry cycles. 

In this study, we unambiguously identified the 29 bacterial species listed in Table 1. Shown are the substrate material from which the sample was taken, gram status, the most likely source, and the characterization method used for identification of the bacterial species. While the obtained results were found to be independent of brand and model of the washing machines, nature and quantity of the identified microorganisms varied when comparing the different material categories, in good agreement with previous studies [24].

### 3.2. Analysis of Biofilm Formation

Given their abundance in all the tested samples in this and previous studies, we selected all the pseudomonads and staphylococci listed in Table 1 for culturing and further analysis. Pseudomonads are commonly found in tap water, whereas staphylococci are typically introduced by the dirty load, rendering both bacterial classes suitable candidates for a potential biofilm model system. In addition, two laboratory strain bacteria were included in the study, *Escherichia coli* XL1-blue and *Pseudomonas putida*. *E. coli* typically colonizes the lower intestine of humans and may be introduced by the laundry. *Pseudomonas putida* was previously described as a strong biofilm producer in domestic appliances [6]. 

For the analysis and comparison of the biofilm-forming potential of each bacterial monoculture, we used standard quantitative biofilm assays, such as the indirect quantification method crystal violet staining and Congo red agar (see Appendix A), as well as a direct quantification method determining viable cell numbers by plate count (colony-forming unit assay). We additionally tested if different culture media have an effect on biofilm formation. Firstly, we chose the Luria broth (LB) as a full medium to analyze standard biofilm formation, secondly, the brain heart infusion (BHI) as a richer full medium to analyze whether a richer medium builds a more pronounced biofilm. Table 2 provides an overview of the obtained results using the various methods. 

The biofilm-forming potential of our isolated bacteria from domestic appliances was tested on the Congo red agar. This method is a standard method for detection of biofilm-forming staphylococci relying on their ability to cleave sugar. In Appendix A, the color ranking of the CRA is shown. The black colonies with crystalline appearance are considered high producers, dark red—moderate producers, red-pink colonies—biofilm non-producers. With the Congo red agar method, as shown in Appendix A, all staphylococci are strong biofilm producers. None of the tested pseudomonads were able to produce black colonies. 

We evaluated monoculture biofilm formation on U-bottom (hydrophobic) and F-bottom (hydrophilic) polystyrene plates using BHI and LB media. The plates were measured at 595 nm and the calculated value is the average of three wells. Figure 1 shows the monoculture biofilm formation results for the pseudomonads and staphylococci listed in Table 1.

To verify the information of the manufacturers regarding surface hydrophobicity of the used plates, we measured the water contact angle with a goniometer. This information helped us to relate the correlation between biofilm formation and wettability of the used surfaces. For contact angles below 90°, the surface is considered hydrophilic. Contact angles larger than 90° are considered hydrophobic [37]. A F-bottom plate with a contact angle of 64.8 ± 2.2° has a hydrophilic surface (Figure 1a) and a U-bottom plate with a contact angle of 90.1 ± 2.4° has a more hydrophobic surface compared to the other plates (Figure 1b). Our results indicate that biofilm formation of most bacterial strains exhibit no dependence on surface material and growth medium. But some species exhibit strong dependence on these factors. We observed a significant difference in monoculture biofilm production by *P. stutzeri* and *S. arlettae* in BHI and LB media cultivated in hydrophilic F-bottom and hydrophobic U-bottom polystyrene plates. While both species prefer the LB medium for biofilm production, *P. stutzeri* prefers a hydrophilic surface, whereas *S. arlettae* grows preferentially on hydrophobic plates. The strongest monoculture biofilm producer in these assays is unambiguously *P. aeruginosa*. In this case, biofilm production seems to be largely independent of the type of growth medium (BHI and LB) and properties of the substrate material (hydrophobic or hydrophilic). Interestingly, *P. putida* (DSMZ) is in the group of low biofilm producers under the culture conditions used in this study. It should be pointed out that the above results strongly depended on the experimental conditions of the assays. Selection of different substrate materials and/or nutrient fluids may yield different results. 

The analysis of biofilm production on glass surfaces was conducted by using conventional laboratory glass tubes. The effect of growth media was analyzed by choosing two different full broth media (BHI and LB). As shown in Figure 2A, crystal violet staining clearly revealed the attachment of some bacteria on glass surfaces for both media. The influence of the media did not seem to have a major effect on the attachment of bacteria to glass. Only *P. mendocina* exhibited significantly higher absorbance in the BHI medium compared to the LB medium. *P. aeroginosa*, *P. mendocina,* and *P. stutzeri* cultivated in the BHI medium revealed significantly higher absorbance values than all other tested bacteria. In Figure 2B, the results are shown for the tested staphylococci and pseudomonads. Here, we experienced that the growth medium has a major effect on biofilm production. Remarkably, *P. aeruginosa* exhibited a significantly higher number of colony-forming units when cultured in the LB medium. *S. haemolyticus* and *P. mendocina* exhibited a significantly higher number of colony-forming units when cultured in the BHI medium. 

### 3.3. Ultrastructural Investigation of Biofilm Formation by Scanning Electron Microscopy (SEM)

Furthermore, we analyzed the biofilm formation process of the strongest biofilm producer *P. aeruginosa* using glass under in vitro conditions (Figure 3). We followed the development of a monoculture biofilm from the initial attachment of bacteria in planktonic form to dispersion (Figure 3A) to glass coverslips. Generally, bacteria have pili (see arrows) which they lose after irreversible adhering to surfaces (Figure 3B). After irreversible attachment, bacteria grow, proliferate, seen as dividing cells with a cell wall in between (for better visualization, the bacteria are colored in blue, Figure 3C), and form a dense EPS matrix containing proteins, eDNA, and polysaccharides (a small EPS area as an example is colored in yellow, Figure 3C). Over time, the biofilm grows horizontally (maturation phase), increases in density and structural complexity, and forms channels (see arrows) for nutrient and water supply (Figure 3D and in higher magnification in Figure 3E). During the maturation phase, microorganisms start to secrete alginate slime (see arrows, colored in green, Figure 3F). The slime protects microorganisms against chemicals, antibiotics, and other environmental threats. In the dispersion phase, a special 3D structure is built, the so-called mushroom-like body (colored in purple, Figure 3G), which can release planktonic bacteria into the aqueous environment (see arrow, one example is colored in blue, Figure 3G). Planktonic bacteria can swim to other locations where more nutrients are available and a new cycle of biofilm formation can start. 

## 4. Discussion

We isolated and cultured bacteria from biofilms of the following surfaces: glass, metal (steel), elastomers (rubber), and plastic. The results of this study are the first critical step towards a more representative biofilm model system (BMS). This might allow the development of more effective antimicrobial materials and surfaces in domestic appliances. Using state-of-the-art MALDI-TOF-MS, a method used by all certified microbiology laboratories for bacterial species determination, we unambiguously identified bacterial species on a variety of materials. The individual strains were isolated and cultivated under laboratory conditions in order to investigate the biofilm formation behavior (monocultures) and its dependence on the nature of the substrate material and the environmental conditions. While the identified microbial communities had been observed previously in domestic appliances, we demonstrated that bacterial isolates from wildtype biofilms show a dominance of pseudomonads from tap water (dominantly isolated from the compartments which are in contact with tap water) and staphylococci introduced by human clothing (dominantly isolated from the compartments which are in contact with the washing load). However, in contrast to previous studies where *Pseudomonas putida* was identified as the primary biofilm producer [6], we did not observe *P. putida* in any of the extracted samples and found *P. aeruginosa* to be the best biofilm producer. The differences in the reported results may originate from the characterization method used for species identification. Our analysis of 16S rRNA sequences (Appendix A) shows a high degree of sequence homology within the pseudomonads, which confuses species identification by analysis of 16S rRNA sequences only. Therefore, we used MALDI-TOF-MS, a method which can easily discriminate between *P. putida* and *P. aeruginosa* [75]. Alternatively, this discrepancy may be explained by differences in cultivation conditions, particularly, temperature, growth medium, and growth rate.

Congo red agar is a standard method to test the ability of biofilm production by staphylococci. Freeman et al. described this method for staphylococci, which is clearly revealed by our results [38]. All the tested staphylococci are detected as positive biofilm producers on the Congo red agar. In contrast, our other assays (such as MTP, TM, and CFU) reveal that only *S. arlettae*, *S. epidermidis*, and *S. haemolyticus* are moderate biofilm producers. This could imply that the staphylococci used in our experiments do not adhere very well to our tested materials. From the six *Staphylococcus* species isolated, *S. haemolyticus* and *S. epidermidis* were the strongest biofilm producers with their own prodigy.

While both MTP and TM assays suggest pseudomonads to be strong biofilm producers, the tested strains showed negative results on the Congo red agar. A possible explanation could be the difference in the composition of the EPS matrix and/or the absence of specific sugar-cleaving enzymes.

*Pseudomonas* species such as *P. aeruginosa* and *P. mendocina* were dominant biofilm producers (monocultures) on all the tested materials (hydrophilic/hydrophobic polystyrene surfaces and glass). Both organisms were able to produce biofilms with life prodigy and were identified as the best biofilm-forming bacteria using crystal violet staining. Interestingly, *P. oleovorans* was identified on all the material samples isolated from naturally grown biofilms, but was found to exhibit only moderate biofilm production under the conditions of our assays.

The species *P. stutzeri* and *S. arlettae* prefer the LB medium for biofilm formation when assayed in cell culture plates and analyzed by crystal violet staining. *P. stutzeri* prefers hydrophilic polystyrene surfaces for biofilm formation, whereas *S. arlettae* grows preferably on hydrophobic polystyrene surfaces. The lipopolysaccharide of gram-negative bacteria (*P. stutzeri*) is highly charged (negative) [76]. For gram-positive bacteria like staphylococci, cell wall macromolecules are responsible for adhesion to hydrophobic surfaces [77,78]. The number of contact-forming macromolecules therefore defines the adhesive strength of staphylococci like *S. arlettae*. In real life, it might be much more complicated, since not all gram-negative bacteria show preference to hydrophilic surfaces [79]. The life prodigy of *P. aeruginosa* biofilms shows remarkably higher results in the LB medium. One possible explanation could be strong adherence of *P. aeruginosa* to glass. Compared with the crystal violet staining results in glass tubes and on culture plates, biofilm formation is independent of the media. Another reason for the different results indicating biofilm formation by MTP, TM, and CFU assays might also be that crystal violet staining probes all components of the biofilm (i.e., dead cells, proteins, RNA, cDNA, polysaccharides, etc.), whereas the CFU method only reveals the number of viable cells detached from the glass surface. Given these differences, results from crystal violet staining may not correlate with the high number of viable cells. However, our results show that biofilm formation depends more on bacterial species and the material than on the used medium. Contrary to expectations, a full rich medium such as the BHI does not correlate with strong biofilm formation. 

By observing different maturation stages of the *P. aeruginosa* biofilm using SEM, a better understanding of the biofilm lifecycle was gained. Biofilm formation of *P. aeruginosa* has been extensively studied [80,81,82,83,84,85]. Nevertheless, most of the studies do not analyze different maturation stages and therefore no biofilm lifecycle of *P. aeruginosa* [80,81,82,83,84]. To the best of our knowledge, only one study is published analyzing biofilm formation of *P. aeruginosa* over time [85]. In contrast to our study, all the studies cited above used strains from culture collections and no freshly isolated strains from extreme habitats like domestic appliances. We were able to reveal that this difference is also clearly reflected by the timeframe of biofilm formation. Carette et al. showed the dispersion phase after 144 h, whereas our freshly isolated *P. aeruginosa* reached this phase after as little as 72 h [85]. 

In contrast to the previously published studies concerning biofilms in washing machines, *P. putida* and *E. coli* were not detected. The laboratory reference strain *P. putida* formed only little detectable biofilm in our assays compared to the other *Pseudomonas* isolates from washing machines. However, this is not surprising, as Gattlen et al. [6] showed that the majority of washing machine isolates formed more biofilms than their ordered reference strains in their study.

We conclude that a biofilm model system and its interaction with different materials and surfaces may be analyzed by a combination of MALDI-TOF-MS as an unambiguous identification method and several standard quantification methods (e.g., MTP, TM, and CFU). In addition, advanced imaging technologies such as SEM can be employed in situ to visualize the structure throughout the biofilm lifecycle. MALDI-TOF-MS enables us to clearly identify bacterial isolates from washing machines, which is a remarkable advantage in comparison to other methods. With our biofilm assays, we were able to analyze which bacteria are best in biofilm production. Our results suggest that *Pseudomonas aeruginosa* is best in producing biofilms, independently of the growth medium and substrate. In addition, SEM is superior for the analyses of all growth phases of biofilm formation. Taken together, our study clearly reveals that a combination of MALDI-TOF-MS, quantification methods (such as MTP, TM, and CFU), and the use of SEM is a powerful tool to study the formation process and evolution of biofilm production in vitro. 

Finally, it should be pointed out that we analyzed laboratory-grown biofilm formation under the conditions described here. A limitation of our study might be the use of laboratory conditions to cultivate the biofilm. It might be that biofilms in washing machines are under high selective pressure and exhibit different growth characteristics. Future studies will investigate the effect of cultivation conditions on biofilm formation mechanisms by tailoring environmental factors such as temperature and growth media. We were able to identify biofilm-forming organisms in their natural habitat; we were able to cultivate these bacteria with specific growth media which conserved their biofilm-producing ability. Furthermore, we were able to identify bacterial spectra by biochemical signatures by MALDI-TOF-MS.

## 5. Conclusions

The development of advanced antimicrobial materials and surfaces requires biofilm model systems that allow for accelerated growth in a laboratory environment. The identification of representative bacterial species and their cultivation under proper chemical and environmental conditions are the core elements of a BMS. Being one of the most accurate methods for bacterial identification, we selected MALDI-TOF-MS to characterize biofilm samples extracted from different material surfaces in domestic washing machines: glass, plastic, rubber, and metal. In contrast to previous studies, we were unable to locate *P. putida* in the collected samples. This discrepancy may be explained by the challenges of bacterial species identification using rRNA given the high homology of 16S rRNA in pseudomonads. From the identified species, *P. aeruginosa* and *P. mendocina* were found to be strong biofilm producers under the cultivation conditions used in this study. Both are important germs in human environments and promising candidate species for the BMS.

Finally, we conclude that the use of MALDI-TOF-MS to identify bacterial spectra together with SEM to visualize the lifecycle phases of biofilm formation might be an ideal combination for the study of biofilms. 

## Figures and Tables

**Figure 1 microorganisms-09-00992-f001:**
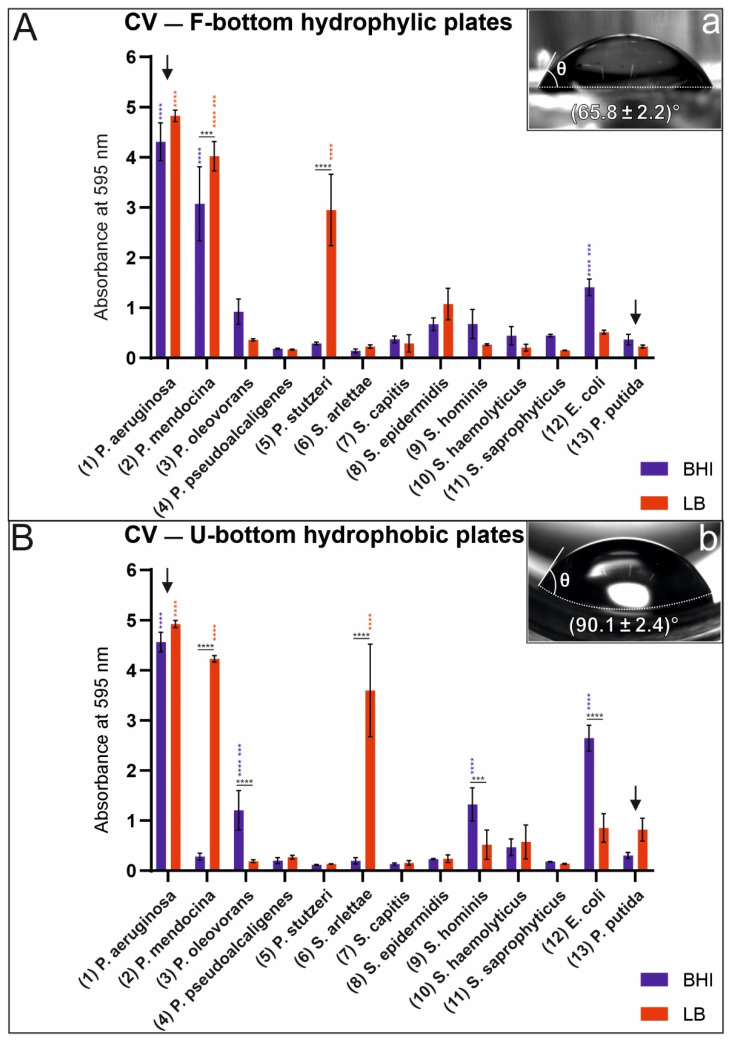
Crystal violet staining of monoculture biofilms using hydrophilic F-bottom (**A**) and hydrophobic U-bottom (**B**) polystyrene plates. Strongest biofilm producers are *P. aeruginosa* (see left arrow) and *P. mendocina*. Goniometer measurements for validation of hydrophilic/hydrophobic surface features of an F-bottom plate with a contact angle of 64.8 ± 2.2° (**a**) and of a U-bottom plate with a contact angle of 90.1 ± 2.4° (**b**). For statistical analysis, a two-way ANOVA or a mixed-effects analysis was performed. Only high-significance values are shown (0.0002 (***), <0.0001 (****)). Black stars in the horizontal direction show the significance values between the media used for cultivation for one bacterial strain. Blue and red stars in the vertical direction indicate the significance values for the BHI medium (blue) and the LB medium (red) based on one bacterial strain compared to all other bacterial strains. Hydrophilic F-bottom plates (**A**): The absorbance values of *P. aeruginosa* and *P. mendocina* cultured in the LB and BHI media are significantly higher than those of every other bacteria. Values of *P. stutzeri* cultured in the LB are significantly higher than those of every other bacteria in respect to *P. aeruginosa* and *P. mendocina*. *E. coli* cultured in the BHI reveals significantly higher absorbance values against all other bacteria in respect to *P. aeruginosa* and *P. mendocina*. Hydrophobic U-bottom plates (**B**): *P. aeruginosa* cultured in the LB and BHI, *P. mendocina* and *S. arlettae* cultured in the LB reveal significantly higher absorbance values than all other bacteria. The absorbance values of *E. coli*, *S. hominis* and *P. oleovorans* cultured in the BHI are significantly higher than those of all other bacteria in respect to *P. aeruginosa. P. putida*, which was described as a strong biofilm producer in previous studies, shows low biofilm formation in these assays (see right arrow).

**Figure 2 microorganisms-09-00992-f002:**
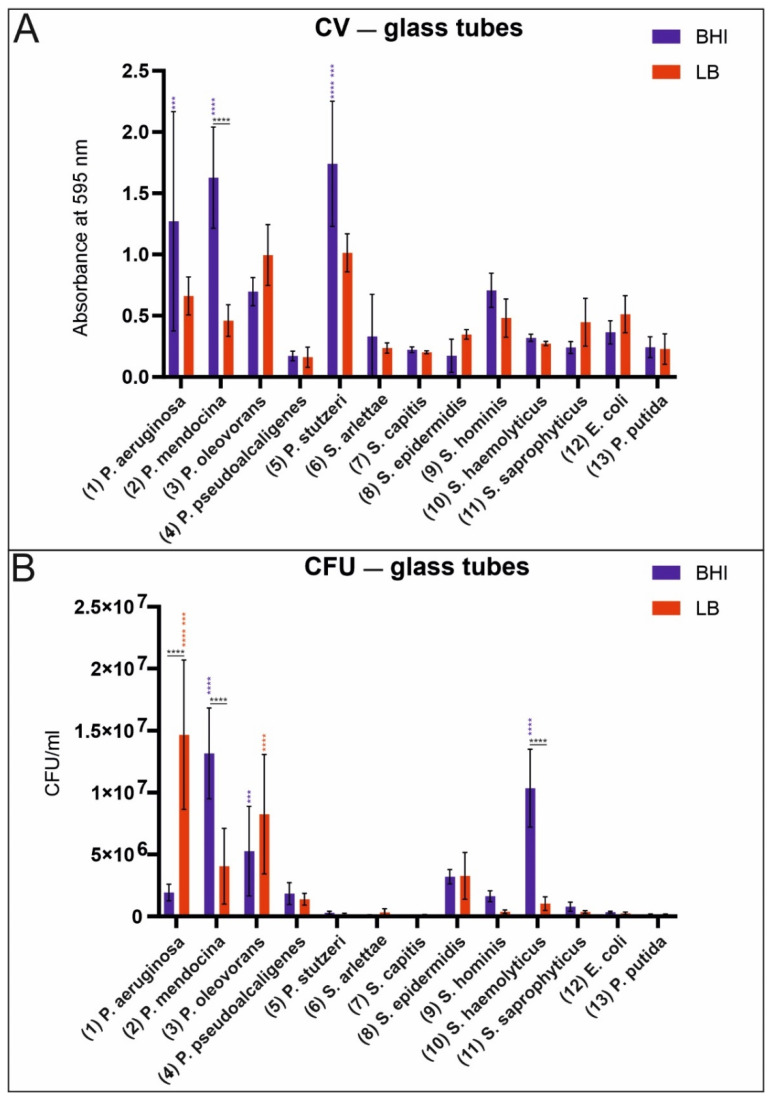
Bacterial attachment/biofilm production in laboratory glass tubes using BHI and LB growth media. Absorbance values of crystal violet staining of biofilms grown in BHI and LB media (**A**) compared to the colony-forming units (CFU/mL) (**B**). For statistical analysis, a two-way ANOVA or a mixed-effects analysis was performed. Only high-significance values are shown (0.0002 (***), <0.0001 (****)). Black stars in the horizontal direction show the significance values between the media used for cultivation for one bacterial strain. Blue and red stars in the vertical direction indicate the significance values for the BHI medium (blue) and the LB medium (red) based on one bacterial strain compared to all other bacterial strains. CV—glass tubes (**A**): The absorbance values of *P. aeruginosa, P. mendocina*, and *P. stutzeri* cultured in the BHI medium are significantly higher than those of all the other bacteria tested. CFU—glass tubes (**B**): The number of colony-forming units of *P. mendocina* and *S. haemolyticus* cultured in the BHI medium is significantly higher than that of all the other tested bacteria. *P. oleovorans* cultured in the BHI medium exhibited significantly more colony-forming units than *S. capitis*. The number of colony-forming units of *P. aeruginosa* and *P. olevorans* cultured in the LB medium is significant greater than that of all the other bacteria.

**Figure 3 microorganisms-09-00992-f003:**
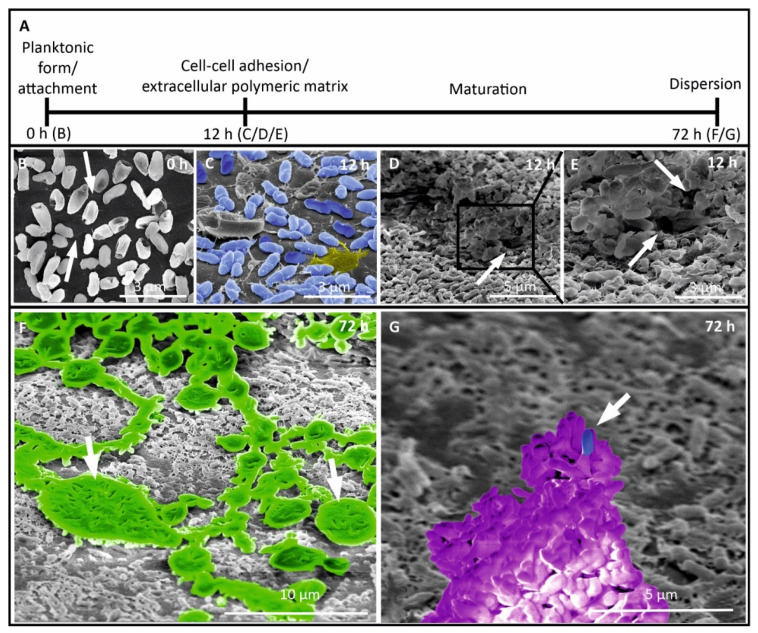
Scanning electron microscopy (SEM) of biofilm production by *Pseudomonas aeruginosa*. (**A**) Schematic representation of biofilm development on surfaces. (**B**) Planktonic form of *Pseudomonas aeruginosa* with pili in place (arrow heads). (**C**) Attached microorganisms on the surface start to grow and cells adhere to the surface (for better visualization, the bacteria are colored in blue), secretion of the exopolysaccharide matrix is starting (a small EPS area as an example is colored in yellow). (**D**,**E**) Bacteria start to form microcolonies with specific coaggregation between them, maturation of the biofilm is characterized by three-dimensional forms such as columns, towers, and channels (arrow heads). (**F**) Ongoing maturation phase shown by firm slime production (arrow heads and colored in green). (**G**) Last phase of biofilm formation, a mushroom-like body (colored in purple) and dispersion of single bacteria (one example is colored in blue) are visualized.

**Table 1 microorganisms-09-00992-t001:** List of bacteria isolated from different material surfaces (material category), including gram status, most likely source, and characterization method used for identification.

Bacteria	Substrate Material	Gram Status	Occurrence/Source	Characterization Method	Score
*Achromobacter piechaudii*	Plastic	-	Soil and water [41]	MALDI-TOF-MS	2.09
*Achromobacter spanius*	Plastic	-	Human blood [42]	MALDI-TOF-MS	2.03
*Acinetobacter lwoffii*	Glass	-	Normal flora of the oropharynx and skin [43]	MALDI-TOF-MS	2.28
*Acinetobacter towneri*	Elastomer	-	Activated sludge, Australia [44,45]	MALDI-TOF-MS	2.29
*Aerococcus viridans*	Plastic, metal, elastomer	+	Hospital environments and room air [46]	MALDI-TOF-MS	2.06
*Bacillus circulans*	Plastic	+	Soil, marine water, plants, animals [47]	MALDI-TOF-MS	2.11
*Bacillus pumilus*	Glass	+	Soil [48];	Classical microbiology	-
*Brevibacterium celere*	Glass	+	Alga *Fucus evanescens* [49]	Classical microbiology	-
*Chryseobacterium shandongense*	Elastomer	-	Soil [50]	MALDI-TOF-MS	2.05
*Delftia lacustris*	Plastic	-	Mesotrophic lake water [51]	MALDI-TOF-MS	2.41
*Kocuria rhizophila*	Elastomer	+	Soil [52], rhizosphere of *Typha angustifolia* [53]	MALDI-TOF-MS	2.31
*Micrococcus luteus*	Elastomer	+	Human skin [54], air [55]	MALDI-TOF-MS	2.45
*Moraxella osloensis*	Elastomer	-	Environmental sources in hospitals and normal human respiratory tract [56]	MALDI-TOF-MS	2.12
*Pantoea septica*	Metal	-	Environment, plants, seeds, vegetables, human skin [57]	MALDI-TOF-MS	2.17
*Pseudomonas aeruginosa*	Plastic, elastomer	-	Water and soil [58]	MALDI-TOF-MS	2.51
*Pseudomonas mendocina*	Plastic, elastomer	-	Water and soil [59]	MALDI-TOF-MS	2.23
*Pseudomonas oleovorans*	Plastic, metal, glass, elastomer	-	Cutting fluid [60]	MALDI-TOF-MS	2.22
*Pseudomonas pseudoalcaligenes*	Plastic	-	Cutting fluid [60,61]	MALDI-TOF-MS	2.09
*Pseudomonas stutzeri*	Glass, elastomer	-	Denitrifying bacteria widely distributed in the environment [62]	MALDI-TOF-MS	2.30
*Raoultella planticola (Synonym: Klebsiella planticola)*	Plastic	-	Radishroot, water, [63,64,65]	MALDI-TOF-MS	2.29
*Serratia liquefaciens*	Plastic	-	River water, domestic sewage, fish [66]	MALDI-TOF-MS	2.03
*Serratia marcescens*	Plastic	-	Water [67]	MALDI-TOF-MS	2.17
*Staphylococcus arlettae*	Metal	+	Skin of mammals and birds [68]	Classical microbiology	-
*Staphylococcus capitis*	Metal	+	Human skin [69]	MALDI-TOF-MS	2.09
*Staphylococcus epidermidis*	Plastic, glass	+	Skin [70]	MALDI-TOF-MS	2.14
*Staphylococcus hominis*	Metal, glass, elastomer	+	Skin [71]	MALDI-TOF-MS	2.26
*Staphylococcus haemolyticus*	Metal, glass, elastomer	+	Skin [72]	MALDI-TOF-MS	2.23
*Staphylococcus saprophyticus*	Elastomer	+	Perineum, rectum urethra, cervix, and gastrointestinal tract of humans, pigs and cows [73]	MALDI-TOF-MS	2.17
*Streptococcus salivarius*	Glass	+	Human oral cavity [74]	MALDI-TOF-MS	2.24

**Table 2 microorganisms-09-00992-t002:** Overview of monoculture biofilm formation behavior of different bacterial species using crystal violet (CV) staining on hydrophilic F-bottom polystyrene cell culture plates (CCP), hydrophobic U-bottom 8-strip plates, and glass tubes. Glass tubes were also analyzed by colony-forming unit (CFU) measurement, the average value is shown in CFU/mL. Biofilm formation was conducted with both the brain heart infusion (BHI) medium and on Luria broth (LB) plates.

Bacteria	CV Absorbance, F-Bottom Polystyrene Plates (Hydrophilic)		CV Absorbance, U-Bottom Polystyrene Plates (Hydrophobic)		CFU, Glass Tubes (CFU/mL)		CV Absorbance, Glass Tubes	
	BHI	LB	BHI	LB	BHI	LB	BHI	LB
(1) *P. aeruginosa*	4.30	4.82	4.56	4.92	1.93 × 10^6^	1.47 × 10^7^	1.27	0.66
(2) *P. mendocina*	3.07	4.02	0.28	4.22	1.32 × 10^7^	4.05 × 10^6^	1.63	0.46
(3) *P. oleovorans*	0.92	0.36	1.20	0.19	5.27 × 10^6^	8.27 × 10^6^	0.70	1.00
(4) *P. pseudoalcaligenes*	0.18	0.16	0.20	0.26	1.85 × 10^6^	1.39 × 10^6^	0.17	0.16
(5) *P. stutzeri*	0.29	2.94	0.11	0.13	2.88 × 10^5^	1.67 × 10^5^	1.74	1.01
(6) *S. arlettae*	0.14	0.22	0.19	3.59	8.33 × 10^4^	3.35 × 10^5^	0.33	0.24
(7) *S. capitis*	0.37	0.29	0.13	0.15	4.86 × 10^4^	8.80 × 10^4^	0.22	0.20
(8) *S. epidermidis*	0.67	1.07	0.23	0.23	3.20 × 10^6^	3.27 × 10^6^	0.17	0.35
(9) *S. hominis*	0.67	0.263	1.32	0.51	1.64 × 10^6^	3.78 × 10^5^	0.71	0.48
(10) *S. haemolyticus*	0.44	0.20	0.46	0.57	1.04 × 10^7^	1.03 × 10^6^	0.32	0.27
(11) *S. saprophyticus*	0.44	0.15	0.18	0.13	7.85 × 10^5^	3.63 × 10^5^	0.24	0.45
**Comparative Bacteria**								
(12) *E. coli* XL1-blue	1.40	0.15	2.64	0.85	3.53 × 10^5^	2.50 × 10^5^	0.36	0.51
(13) *P. putida*	0.36	0.22	0.30	0.81	1.38 × 10^5^	1.72× 10^5^	0.24	0.23

## Data Availability

The 16S rRNA Data used for this study can be found in the PubMed library. All presented data are included in the manuscript and Appendix A.

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
