# Peer review of "Identification of Microorganisms from Several Surfaces by MALDI-TOF MS: P. aeruginosa Is Leading in Biofilm Formation"

_microorganisms, 2021, doi:10.3390/microorganisms9050992_

Round 1

Reviewer 1 Report

Why did authors removed congo red assay, it added compliment to MALDItof and SEM. Congo red helps in screening strong and weak biofilm producer. It can be added as supplementary data.

P. aeruginosa should be italized, please double check through out manuscript.

Y axis tile "Absorbance at 595nm" remove [nm].

Reviewer 2 Report

I am content with the corrections made and overall improvement of the paper. 

Author Response

We thank the reviewer for being happy  with our manuscript. 

This manuscript is a resubmission of an earlier submission. The following is a list of the peer review reports and author responses from that submission.

Round 1

Reviewer 1 Report

In this study authors aimed to use MALDI-TOF-MS with SEM to identify bacterial species from environmental biofilms that can be used in a model biofilm system.

  • In the first paragraph of the introduction authors reference (Vilanova et al., 2015) as a test citation rather than as a number. Please ensure reference style is consistent throughout.
  • I would expect that anyone reading this paper would have some foundational knowledge on biofilm definitions and the main stages of biofilm formation. Therefore, I do not think that the paragraphs in the introduction on these subjects are necessary.
  • From reading the manuscript, I am unsure of why the two growth mediums were chosen. The authors should provide more detailed explanation.
  • Why were no statistical analyses performed on biofilm CFUs or absorbance values?
  • The manuscript is titled as using MALD-TOF-MS and SEM to analyse biofilm formation but the methods describe multiple other methods such as CFU enumeration and Crystal Violet staining which are two well characterised methods. The methods listed in the title appear to be used for identification and not for analysis. I would suggest a change of title to reflect this.
  • I am not sure why so many biofilm analysis methods are needed (CFU, congo red agar, microtitre plate and tube method). Authors also report that bacterial biofilms were found on rubber and metal substrates. Surely these substrates should also be used in the development of the model system?
  • Bacterial biofilm formation has been shown to be a highly heterogeneous mechanism therefore I do not think classifying biofilm formation into ‘high’, ‘moderate’ and ‘weak’ as done in table 2 is accurate enough.
  • The authors state that no Pseudomonads were classified as biofilm producers but the results say otherwise (P. aeruginosa and P. mendocina appear to form robust biofilms). Authors also make this claim based on findings in table 3. I cannot see a third table.
  • Figure 1 does not contribute to the manuscript. This would be better as a supplementary figure.
  • Figure 4 does not contribute to scientific knowledge. Formation of P. aeruginosa biofilms has been extensively studied and figure 4 seems to repeat previous findings of biofilm structures.
  • Despite SEM being mentioned in the title, it only features once in the manuscript. I also struggle to see how this can be used as a method to analyse biofilm formation as mentioned in the title, as it cannot be quantified making analysis too ambiguous.
  • Figure 5 is not mentioned in the text so I am not sure how it contributes to the manuscript.

Reviewer 2 Report

In general, the study looking into bacterial species existence in domestic appliances is of interest to common people. Overall the manuscript is written well and results/figures/Table presented well.

There are few remarks or questions needed to be addressed before consideration for publication, which will improve quality and more clear data.

Methods:

2.1. sample collections comparative strains are they are ATCC isolates? please provide details where they are collected/obtained from.

2.2 Matrix used for MALDI TOF - please explain what type of matrix used, details name (material). to note: even though its clinical practice to analysis sample using MALDITOF -but not all lab has facilities so more information on technique and materials will be useful for young researchers. if needed methods can be elaborated in supplementary section.

2.2 classical microbiology ? what are they...what researchers used in this study.....like isolation agar? etc. please detail

2.3.1 . hydrophilic and hydrophobic - what are the contact angles value...atleast provide values using water measuring using goniometer.

2.3.3. bacteria/biofilm grown for 24 hours 37 celcius..is that in static condition or dynamic (shaking incubator)? if Yes what RPM used?

Biofilms are considered weak or strong based on 0.3 1.0..what are they? absorbance value at 595nm? or just scoring by visual observation..please specify.

Results

3.1 sounds more of a methods/techniques section rather than results. please rearrange it.

3.2 what strains of pseudomonas and staphylococci researchers used in this study..please write clear and full name.

3.2 below figure 1. Please comment why p. stutzeri prefers hydrophilic surface and S. arlettae grows preferably on hydrophobic surface?

Figure 2. Crystal violet absorbance values are very high at 595 nm up to 5...can your plate reader reads that's high absorbance. Generally after 3 the system shows overvalue after absorbance 3.

SEM images...what the time frame for maturation images?

Discussion. Gattlen et al.  include reference. number.

Reviewer 3 Report

The authors conducted a set of assays regarding the biofilm formation on the washing machine surfaces. They analyzed the strains with MALDI-TOF MS, quantified with CRA, MTP, TM and CFU, and used SEM for studying the formation, processes and evolution of biofilm production in vitro. The study they performed is interesting, novel and of significance regarding the biofilm issues with everpresent household appliances of low ecological footprint.

Some minor issues:

  1. How many biological replicates were used for each strain for MALDI-TOF MS analyses? Please state in the section 2.2.
  2. Were the MALDI-TOF MS measurements conducted in duplicate/triplicate? Please state in the section 2.2.
  3. Indicate the identification scores of MALDI-TOF MS analyses for each strain in the table 1. Clarify the classification of the scores in the section 2.2.
  4. in "Droplets roll down the leave..." replace leave with leaf
  5. in "...real-live environments..." replace live with life
  6. in "bacteria" species, cultivation, strains, replace bacteria with bacterial
  7. in section 2.2. is it five different swabbing locations or six as stated above?
  8. rephrase the sentence "Forevermore subsequently......"
  9. instead of tripled use "in triplicate"
  10. instead of "The average of three wells..." use "The average score of...."
  11. instead of ml use mL
  12. instead of "pub med" use "PubMed"
  13. In Figure 1 caption use biofilm in lowercase
  14. in "...the above results are strongly depended.." omit "are"
  15. mushroom-like
  16. "....we unambiguously identified...."
  17. instead of Fig 6. use Fig 5. in text
  18. unify font in Discussion
  19. instead of Gattlen et al. use numeral (6)?
  20. italicize in situ, in vitro
  21. P.putida, not P.Putida
  22. Pseudomonads, not Pseudomonades
